# *Plasmodium falciparum* infection in humans and mosquitoes influence natural Anopheline biting behavior and transmission

Christine F. Markwalter [1,9], Zena Lapp[1,9], Lucy Abel[2], Emmah Kimachas[2], Evans Omollo[3], Elizabeth Freedman[4], Tabitha Chepkwony[2], Mark Amunga[2], Tyler McCormick[5], Sophie Bérubé [6], Judith N. Mangeni[7], Amy Wesolowski [6], Andrew A. Obala[8], Steve M. Taylor [1,4] ✉ & Wendy Prudhomme O'Meara [1,4] ✉

The human infectious reservoir of *Plasmodium falciparum* is governed by transmission efficiency during vector-human contact and mosquito biting preferences. Understanding biting bias in a natural setting can help target interventions to interrupt transmission. In a 15-month cohort in western Kenya, we detected *P. falciparum* in indoor-resting *Anopheles* and human blood samples by qPCR and matched mosquito bloodmeals to cohort participants using short-tandem repeat genotyping. Using risk factor analyses and discrete choice models, we assessed mosquito biting behavior with respect to parasite transmission. Biting was highly unequal; 20% of people received 86% of bites. Biting rates were higher on males (biting rate ratio (BRR): 1.68; CI: 1.28–2.19), children 5–15 years (BRR: 1.49; CI: 1.13–1.98), and *P. falciparum*-infected individuals (BRR: 1.25; CI: 1.01–1.55). In aggregate, *P. falciparum*-infected school-age (5–15 years) boys accounted for 50% of bites potentially leading to onward transmission and had an entomological inoculation rate 6.4x higher than any other group. Additionally, infectious mosquitoes were nearly 3x more likely than non-infectious mosquitoes to bite *P. falciparum*-infected individuals (relative risk ratio 2.76, 95% CI 1.65–4.61). Thus, persistent *P. falciparum* transmission was characterized by disproportionate onward transmission from school-age boys and by the preference of infected mosquitoes to feed upon infected people.

*Plasmodium falciparum* parasites are transmitted between human host and mosquito vector to maintain a complex and robust transmission cycle. The number of human *P. falciparum* infections arising from a single infected person ($R_O$) is governed by the rate of vector-host contact and by the efficiency of parasite transmission during contact[1]. Therefore, both nonrandom vector-host contact and transmission efficiency can drastically alter transmission patterns and influence intervention effectiveness[2]. To accurately estimate transmission patterns and efficiently deploy transmission-reducing interventions, it is important to understand who is being bitten by *P. falciparum* vectors, who is transmitting to mosquitoes, and what mosquito characteristics shape biting behavior.

[1]Duke Global Health Institute, Duke University, Durham, NC, USA. [2]Academic Model Providing Access to Healthcare, Moi Teaching and Referral Hospital, Eldoret, Kenya. [3]Duke Global Inc, Nairobi, Kenya. [4]Division of Infectious Diseases, Duke University School of Medicine, Durham, NC, USA. [5]Departments of Statistics & Sociology, University of Washington, Seattle, WA, USA. [6]Department of Epidemiology, Johns Hopkins Bloomberg School of Public Health, Baltimore, MD, USA. [7]School of Public Health, College of Health Sciences, Moi University, Eldoret, Kenya. [8]School of Medicine, College of Health Sciences, Moi University, Eldoret, Kenya. [9]These authors contributed equally: Christine F. Markwalter, Zena Lapp. ✉e-mail: steve.taylor@duke.edu; wpo@duke.edu

Contacts between humans and mosquitoes are driven by mosquito vector abundance and biting preferences[3]. Laboratory and semi-field studies have demonstrated high inter-individual variability in attractiveness to mosquitoes[4,5] and that *Plasmodium* infection in the mosquito influences mosquito biting preferences and behaviors[6,7]. Studies of human and mosquito samples collected in natural settings have also demonstrated strong heterogeneity in mosquito bite exposure[8], as well as in which people carry gametocytes and are likely to transmit to mosquitoes[9–12]. While these works have provided valuable insight into the human *P. falciparum* infectious reservoir, they are experimental[10–16], cross-sectional[8,17–19], short-term[20], or semi-longitudinal studies[21]. Few studies quantify variation in mosquito-human contact in a natural setting across a well-defined host population over an entire transmission season, and, to our knowledge, none have incorporated multisource bloodmeals, which can lead to increased biting rates and amplified transmission[22].

Here, we explored mosquito biting behavior and human-to-mosquito transmission in a community-based cohort in Western Kenya using weekly collections of indoor-resting mosquitoes from 75 households over 15 months of observation. To precisely understand who was bitten by an individual mosquito, we matched human short tandem repeat (STR) genotypes between single- and multi-source mosquito bloodmeals and cohort participants. We hypothesized that mosquitoes bite certain groups of people more than others, and that mosquito characteristics, including species and *P. falciparum* infection status, influence their biting behavior.

## Results

From July 2020 to September 2021, we collected 3660 female *Anopheles* mosquitoes across 75 households in 5 villages (Fig. S1). Most mosquitoes (3051, 83%) were collected during the high-transmission season in 2021 (March-August). Of the 3502 mosquitoes for which we could assign species the species, most were *An. gambiae* (1611 *An. gambiae s.s.*, 46%; 130 *An. gambiae s.l.*, 4%) or *An. funestus* (1600, 46%). In 2021, *An. gambiae s.s.* counts peaked slightly before *An. funestus* counts (Fig. S1A). Most mosquitoes (3038) were immediately processed to determine abdominal status (freshly fed, half-gravid, gravid, or unfed) and preserve the blood from freshly fed mosquitoes (1563) to match to cohort participants. The other 622 were reared to enable identification of successful abdominal infections. We also collected 6414 human DBS through active (monthly) and passive case detection, of which 1758 (27%) were positive for *P. falciparum* (Fig. S2A). Individuals were infected at a median of 20% of monthly visits, with little variation across age and gender (Fig. S2B). However, bed net use did vary, with males aged 5–15 reporting the lowest bed net use (Fig. S2C).

### Female Anophelines rest where they fed

Among the 1563 freshly fed female Anophelines, 1064 were available for STR typing (Fig. S3). Of these, 777 (73%) returned STR profiles (Figs. 1A and S3), among which 654 (84%) were single-source bloodmeals, and 123 (16%) had more than one contributor (i.e. multi-source). We identified at least one cohort participant in 662 (85%) bloodmeals (*n* = 550 single-source, *n* = 112 multisource), and an additional cohort participant in 58/112 (52%) matched multi-source bloodmeals (Fig. 1B). Among the 720 identified mosquito-human pairs, the mosquito was collected from the family compound where the person lived in 94% (*n* = 676) of cases (Fig. 1C). Additionally, for the 58 multisource bloodmeals where two participants were matched, the two people were from the same family compound in 88% (*n* = 51) of cases (Fig. 1D).

### Some *P. falciparum* haplotypes in reared mosquito abdomens spatiotemporally match those from infected humans

Among the 622 female *Anopheles* that were reared following collection, 412 survived to day 6 or later, of which 34 (8.3%) harbored

*P. falciparum* in their abdomen. We estimated that the maximum transmission efficiency, defined as the probability of a mosquito successfully becoming infected after biting an infected individual, was 0.16 (95% CI: 0.12–0.22). Of the infected mosquitoes, 30 (88%) infections were successfully sequenced at the *Pf-csp* locus (Fig. S4A). We observed 28 distinct haplotype sequences in mosquitoes, 10 (36%) of which were found in at least one *An. gambiae s.s.* and one *An. funestus* abdomen (Fig. S4B). Additionally, we successfully sequenced 37/57 (65%) *P. falciparum* positive human DBS from household-days that yielded a reared mosquito with a sequenced infection (Fig. S4C). For 13/30 (43%) infected mosquitoes, we identified at least one haplotype match to an infected household member on the household-day of mosquito collection (Fig. S4D). Given this small sample size, we used the STR profile matches for all subsequent analyses.

### Mosquito biting is nonrandom

We next investigated the distribution of mosquito bites across the cohort and which participants had the highest rate of biting. For 382/588 (65%) participants, we did not observe any bites. Among the 206 participants who were bitten, the maximum number of bites observed for an individual in a single night was 12 (median = 1). Across all collections, mean nightly biting rates ranged from 0.03 to 1.5 bites per person-night at risk. Individual biting rates were more unequal (Gini index: 0.82, 95% CI: 0.80–0.85) than expected by random chance (Gini index: 0.51, 95% CI: 0.48–0.54), with 20% of people receiving 86% of bites per night at risk (Fig. 2A, B). The highest biting rates were observed for males 5–15 years old who were infected with *P. falciparum* and did not sleep under a bed net, with an estimated 13.2 bites per month per person (range: 0–210; Fig. 2C). These trends held in a multilevel multivariable model (Fig. 2D, Table 1, and Table S1): we observed higher biting rates on males (biting rate ratio (BRR): 1.68; CI: 1.28–2.19), children 5–15 (BRR: 1.49; CI: 1.13–1.98), and individuals infected with *P. falciparum* (BRR: 1.25; CI: 1.01–1.55), while lower biting rates were observed on net users (BRR: 0.51; CI: 0.40–0.65). All of these trends held in sensitivity analyses, with some wider confidence intervals due to smaller sample sizes (Table S2).

To estimate biting rates on potentially infectious individuals and thereby better understand how the observed differential biting by mosquitoes could influence human contributions to onward transmission, we restricted our multivariable model to individuals infected with *P. falciparum*. This model yielded biting rates (Fig. S5A) similar to those observed among all people (Fig. 2D). From this model, we estimated that, in our cohort, males 5–15 receive about 50% of bites that could lead to onward *P. falciparum* transmission to mosquitoes (Fig. 3A).

Next, to better understand biting heterogeneity on the other side of the transmission cycle, from mosquitoes to humans, we investigated biting rates on people by infectious mosquitoes only (Fig. S5B). We observed similar estimates for the BRRs compared to those among all mosquitoes except for individuals infected with *P. falciparum*, who had a higher BRR among infectious mosquitoes (2.26; CI: 1.40–3.66) compared to all mosquitoes. Estimating relative entomological inoculation rates (EIRs) for each category of individuals revealed that, in our cohort, males 5–15 infected with *P. falciparum* have an EIR 6.4x higher than any other group of people (Fig. 3B).

### Mosquito characteristics influence their choice of bloodmeal

We investigated associations between biting behavior and three mosquito characteristics: species, harboring a multi-source bloodmeal, and the presence of sporozoites in the head-thorax. *P. falciparum*-infected *An. funestus* took more multisource bloodmeals compared to uninfected *An. funestus* (Fisher's exact test, *n* = 277, *p* = 0.002), whereas no difference was observed in the proportion of multisource meals taken by infected and uninfected *An. gambiae s.s*

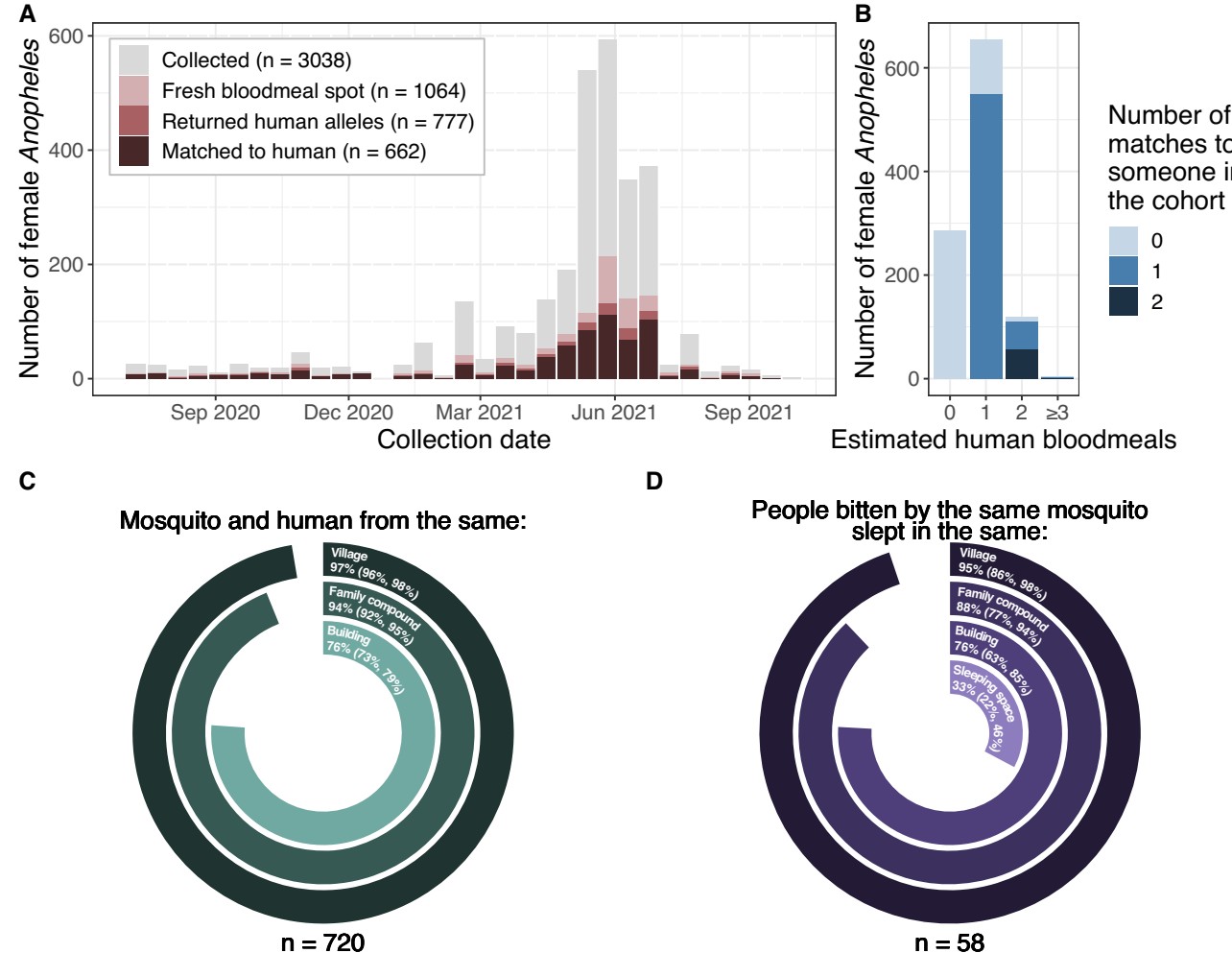

**Fig. 1 | Mosquitoes usually matched to human hosts from the household where they were collected. A** Number of immediately processed mosquitoes (y) by collection date (x), colored by whether they were STR-typed and whether they matched. **B** Matching by number of contributors (NOC, x) to the mosquito bloodmeal. Fill indicates the number of different humans matched to a mosquito bloodmeal. **C** Circle plot indicating whether the mosquito was collected in the same village, family compound, or building as the person it bit. **D** Circle plot indicating whether people that were bitten by the same mosquito came from the same village, family compound, building, or sleeping space. Source data are provided as a Source Data file.

(Fisher's exact test, $n = 287$, $p = 0.439$) (Fig. S6A). However, among mosquitoes positive for *P. falciparum* sporozoites, no differences were observed in sporozoite densities in the head-thoraces between mosquitoes that took single- vs multi-source bloodmeals for both *An. funestus* (Wilcox test, $p = 0.15$) and *An. gambiae s.s.* (Wilcox test, $p = 0.35$) (Fig. S6B).

We next employed a discrete choice model framework to explore how mosquito characteristics impact the relative risk of a bite for humans that differ in each of the risk factors from the previous analysis, assuming a human population similar to the cohort analyzed here (Fig. 4A and Table S3). We observed that host selection was in general similar between species; however, biting preferences varied in relation to number of bloodmeals and mosquito infectiousness.

In contrast with the observations using all bloodmeals, mosquitoes that bit more than one person were less likely to choose to bite a male host (relative risk ratio (RRR): 0.50, 95% CI 0.33–0.77) and more likely to have bitten a child under 5 years (RRR 4.10, 95% CI 1.80–9.35) compared to an individual >15 years. This could be driven, in part, because pairs of people who share a sleeping space are more likely to be found in a multisource bloodmeal compared to people who do not share a sleeping space (unadjusted RR 2.80, 95% CI 1.50–5.23). Females (vs. males) and children <5 years (vs individuals > 15 years) are more

likely to share a sleeping space with someone (unadjusted RR females 1.09, 95% CI 1.00–1.20; unadjusted RR < 5 years 1.23, 95% CI 1.12–1.36).

Amongst the investigated mosquito characteristics, biting behavior was most strongly associated with infection by *P. falciparum* sporozoites. Compared to uninfectious mosquitoes, infectious mosquitoes prefer males over females (RRR 2.02, 95% CI 1.28–3.18), children <5 years (RRR 2.69, 95% CI 1.19–6.09) and 5–15 years (RRR 2.21, 95% CI 1.37–3.59) compared to >15 years, and individuals not sleeping under a net over those who did (RRR 2.22, 95% CI 1.15–4.26). Most strikingly, individuals infected with *P. falciparum* were nearly 3x more likely than uninfected individuals to be bitten by infectious mosquitoes (RRR 2.76, 95% CI 1.65–4.61). This association could result from inherent biases for mosquitoes to bite specific people, or from the presence of non-sporozoite stage *P. falciparum* from a bitten infected person being detected in the head-thorax. Based on the assumption that the contribution of human stage parasites to the measured *P. falciparum* loads in the mosquito head-thoraces would be low, we conducted a sensitivity analysis that tested four different thresholds to classify mosquito head-thoraces with low *P. falciparum* loads as sporozoite-negative. The value of threshold did not change the direction or the significance of the observed effect (Table S4). To further assess the association between the presence of sporozoites

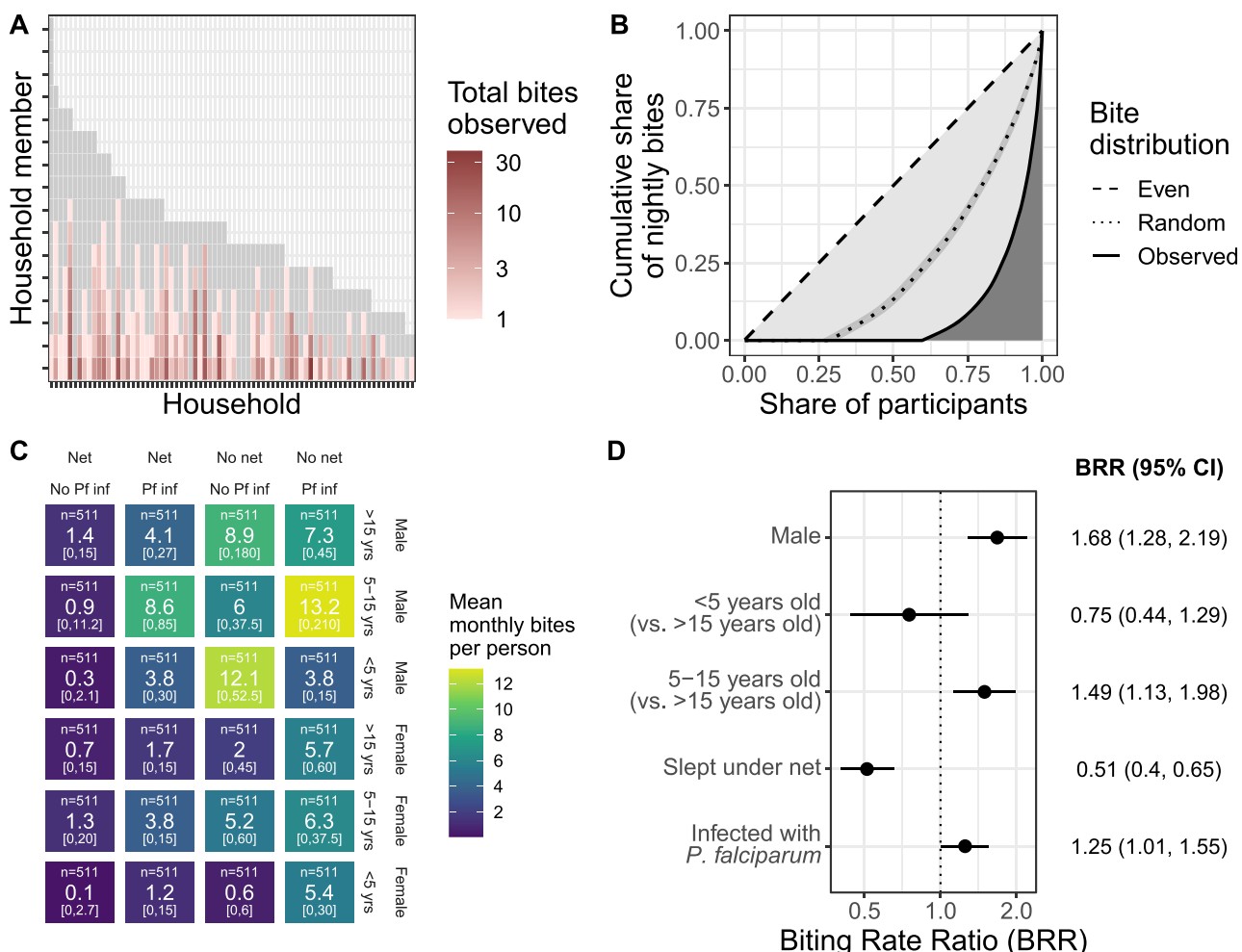

**Fig. 2 | Biting is heterogeneous and nonrandom. A** Individual observed bite counts (color) for each member (y) of each household (x) showing between- and within-household heterogeneity. Gray indicates individuals with no observed bites. Only people present for more than 3 months during the study period were included. **B** The observed distribution of bites is more unequal than random chance. Lorenz curves demonstrating cumulative share of nightly bites (y) across participants (x, sorted by ascending share of bites). Dashed line indicates the null of evenly distributed bites, dotted line the null of simulated (1000x) randomly-distributed bites, and solid line the observed distribution. Overall, 20% of people received 86% of observed bites. **C** Estimated monthly bites per person (from observed average nightly bites; color and number) by 4 risk factors (net use, *P. falciparum* infection status, gender, and age). n indicates the number of people included in the group and the bottom numbers are the range. **D** Biting rate ratios (points) and 95% confidence intervals (lines) for covariates in risk factor analysis based on 2055 person-nights. Adjusters include high transmission season, number of STR-typed mosquitoes in the household, number of household members, whether any household member tested positive for *P. falciparum* by a rapid diagnostic test in the previous month, and number of people in the sleeping space. Source data are provided as a Source Data file.

and a preference to bite infected people, we next tested for a dose-response relationship by repeating the discrete choice model using continuous sporozoite density in place of binary infectiousness. In this model, higher sporozoite densities were associated with a greater likelihood of biting people infected with *P. falciparum* than uninfected people (RRR per 100 sporozoites: 1.92, CI 1.23–2.98; Fig. 4B and Table S5), regardless of species or number of human bloodmeal sources (Fig. S7). This observation further supports the notion that the preference of Anopheline mosquitoes for biting people infected with *P. falciparum* is enhanced by the presence of sporozoites in the mosquito.

## Discussion

In this 15-month longitudinal cohort study in a high-transmission setting in western Kenya, we investigated the human and mosquito factors associated with differential mosquito biting. To do so, we matched human DNA in single- and multi-source *Anopheles* bloodmeals to the individuals they bit and employed measures of inequality, risk factor

analyses, and econometric models of probabilistic choice to assess mosquito biting behavior with respect to both human-to-mosquito transmission and mosquito-to-human transmission. We observed that school-age boys (5–15 years old) were bitten the most, that infectious mosquitoes were more likely to bite participants harboring *P. falciparum* parasites, and that this preference to feed on infected people was enhanced by the presence of higher sporozoite loads in the mosquito head-thorax. Taken together, these results suggest that school-age boys disproportionately contribute to onward *P. falciparum* transmission and that *P. falciparum* sporozoites may modify mosquito biting preferences to favor feeding on infected people.

Infectious mosquitoes were nearly 3x more likely to bite people harboring *P. falciparum* parasites (vs. uninfected people) compared to uninfected mosquitoes. This observation was consistent when expressed as a density of sporozoites, supporting a causal link between the presence of sporozoites and a preference to feed upon infected people. Note that, after a mosquito bites an infected person, it takes about 7–10 days for the mosquito to become infectious; therefore, we

**Table 1 | Person-night characteristics of bitten and not bitten people**

| Characteristic | Overall N = 1789[a] | Bitten N = 398[a] | Not bitten N = 1391[a] | p-value[b] |
|---|---|---|---|---|
| Age | | | | $8 \times 10^{-9}$ |
| >15 (%) | 823 (46) | 156 (39) | 667 (48) | |
| 5–15 (%) | 794 (44) | 225 (57) | 569 (41) | |
| <5 | 172 (9.6) | 17 (4.3) | 155 (11) | |
| Gender | | | | $2 \times 10^{-11}$ |
| Female (%) | 931 (52) | 148 (37) | 783 (56) | |
| Male (%) | 858 (48) | 250 (63) | 569 (44) | |
| Slept under net (%) | 849 (47) | 133 (33) | 716 (51) | $2 \times 10^{-10}$ |
| Infected with P. falciparum | | | | $4 \times 10^{-6}$ |
| Negative (%) | 1111 (62) | 208 (52) | 903 (65) | |
| Positive (%) | 678 (38) | 190 (48) | 488 (35) | |
| Number of people in sleeping space | 3.00 (2.00, 3.00) | 2.00 (2.00, 3.00) | 3.00 (2.00, 3.00) | 0.003 |
| RDT+ household member in prior month (%) | 874 (49) | 211 (53) | 663 (48%) | 0.060 |
| Number of people present in household | 7.00 (6.00, 10.00) | 7.00 (6.00, 9.00) | 7.00 (6.00, 10.00) | $5 \times 10^{-7}$ |
| Number of STR-typed mosquitoes collected in household | 2.0 (1.0, 3.0) | 3.0 (1.0, 6.8) | 1.0 (1.0, 3.0) | $5 \times 10^{-21}$ |
| Transmission season | | | | 0.002 |
| Low (%) | 496 (28) | 86 (22) | 410 (29%) | |
| High (%) | 1293 (72) | 312 (78) | 981 (71%) | |

[a] n (%); median (IQR).
[b] Pearson's $\chi^2$ test; Wilcoxon rank sum test (two-sided).

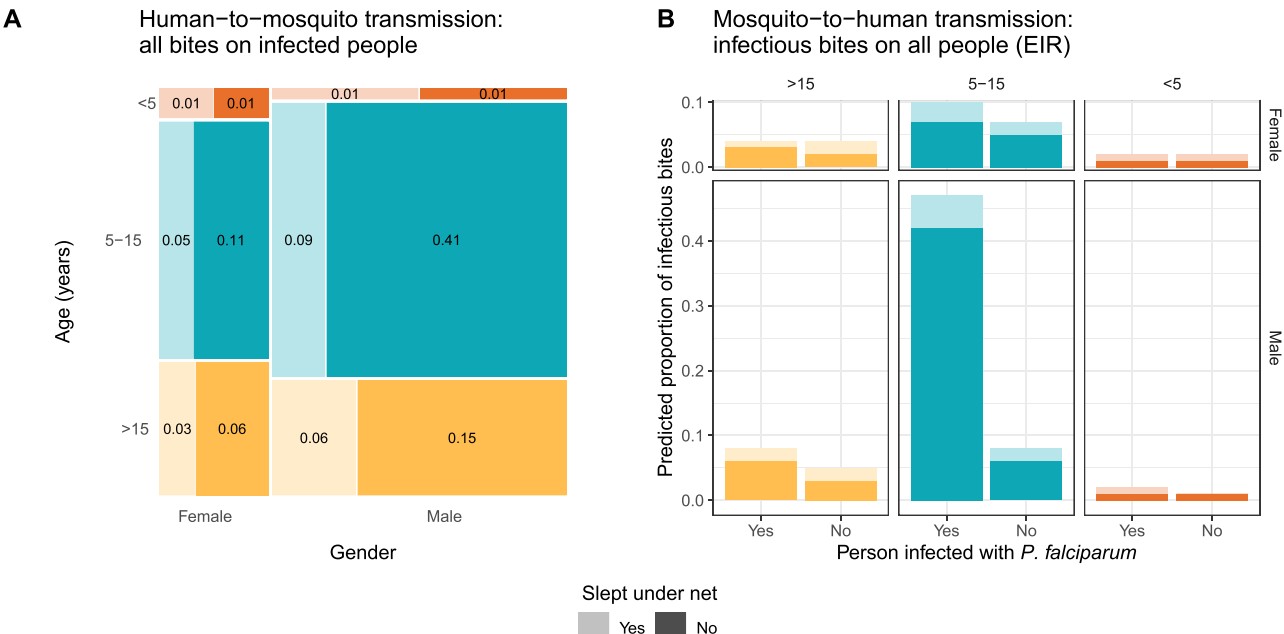

**Fig. 3 | Estimated group-level biting rates with respect to potential *P. falci-parum* transmission. A** Human-to-mosquito *P. falciparum* transmission using a model including only infected individuals. Groups separated by gender on the x-axis, categorical age on the y-axis (and color), and net use by shade. Size of box and number displayed in box show the proportion of bites received by individuals infected with *P. falciparum* in the cohort, representing potential contribution to onward transmission. **B** Mosquito-to-human transmission using a model including only infectious mosquitoes. Color and shade indicate the categories as in panel A. Source data are provided as a Source Data file.

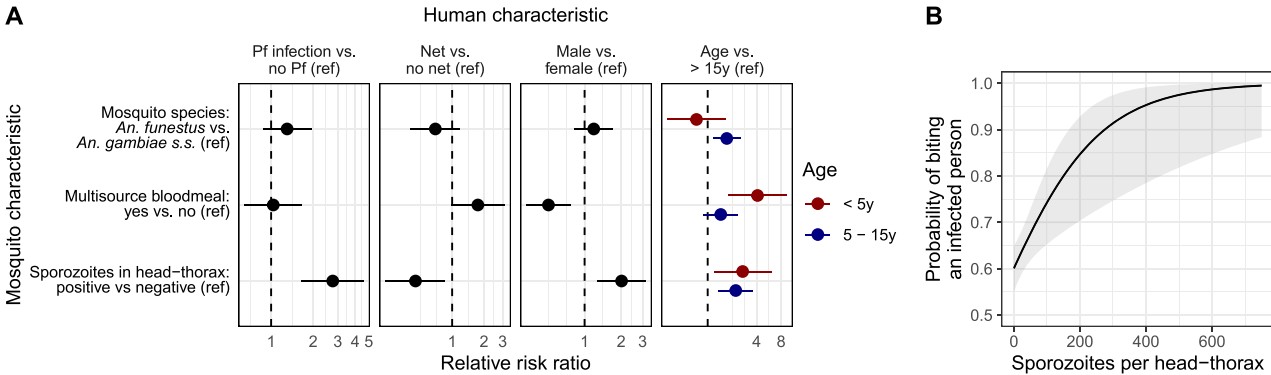

**Fig. 4 | Mosquito characteristics influence their choice of bloodmeal. A** Relative risk ratios estimated in choice models on the preference for humans with various characteristics. A separate multivariable choice model was run for each human characteristic. Dots indicate point estimates, and bars represent the 95% confidence intervals. Models were based on 3516 mosquito-household member pairs. Source data can be found in Table S3. **B** Probability of biting an infected person as a function of *P. falciparum* density. Gray ribbon indicates the 95% confidence interval. Source data are provided as a Source Data file. Ref Reference, y years.

can assign directionality to the relationship between mosquito infectiousness and human *P. falciparum* infection status here. However, the PCR method we used to detect *P. falciparum* in mosquito head-thoraces cannot discriminate between different stages of the parasite; therefore, it important that further studies investigate whether this observation holds when specifically testing for sporozoite-stage *P. falciparum* in the mosquito head-thorax. Even so, to our knowledge, this is the first evidence that *P. falciparum* sporozoites may manipulate mosquito feeding to favor *P. falciparum*-infected humans. Several prior studies have provided experimental evidence that *Plasmodium spp* sporozoite infection modifies specific aspects of Anopheline feeding behavior, including enhancing probing time[23], feeding persistence[24], feeding success[25,26], and attraction to human odors[6]. Prior laboratory and semi-field studies have also reported that being infected with *Plasmodium spp* increases a mouse's[27] or person's attractiveness to mosquitoes[28–30], and our observations suggest that this increased attractiveness is further enhanced when the mosquito is likewise infected with *P. falciparum*. Such enhancement could result if sporozoite infection increases olfactory responsiveness[7,31] to volatiles released during human *P. falciparum* infection. One question that remains unanswered is whether this heightened attractiveness is present among all *P. falciparum*-infected individuals or differs by asexual parasite or gametocyte density. Regardless, how this phenomenon might influence overall transmission intensity is not clear: although modeling suggests that altered mosquito behavior due to *P. falciparum* infection could increase the force of infection by several-fold[32], it is possible that, given the high biting heterogeneity we observed, it may also concentrate the force of infection into a small group of perpetually infected people and therefore not increase population-level prevalence. Either way, our observations support the notion that, in this natural setting, *P. falciparum* parasites may manipulate Anopheline vectors to favor feeding on humans who are already likewise infected with *P. falciparum*.

We estimated biting rates for three distinct groups of mosquito-human pairs: 1) all mosquitoes and all participants, to assess overall mosquito biting preferences; 2) all mosquitoes and people harboring *P. falciparum* parasites, to assess what portion of the human population likely contributes most to transmission to mosquitoes; and 3) infectious mosquitoes and all participants, to assess the relative EIR for each human risk factor. The highest biting rates for all of these groups were on boys aged 5–15 years. This demographic accounted for 50% of all bites potentially leading to parasite transmission to mosquitoes despite comprising only 19.2% of the population at risk. Although individuals sleeping under an insecticide-treated net (ITN) received

fewer bites, school age boys who reported using nets still had very high biting rates when compared to other age groups using nets, possibly due to behavioral differences that make them more accessible during mosquito biting times, such as going to bed late or failing to use an ITN consistently. Previous studies have shown that this age group is also most infective to mosquitoes in feeding assays[10] and comprises the largest segment of the infectious reservoir, defined as infectiousness to mosquitoes projected over a population[10,12,33]. Given that the rate of secondary mosquito infections is a product of an individual's mosquito exposure and their infectiousness to mosquitoes[34], the disproportionate contribution of school-age boys to onward *P. falciparum* transmission reported here is likely underestimated. These observations, coupled with the understanding that school age children more often harbor asymptomatic or chronic infections[35–38], point to them playing a significant role in maintaining transmission and highlight the importance of targeting interventions toward this group as a means to efficiently reduce overall community transmission.

Prior studies using STR typing in wild-caught bloodfed mosquitoes have also investigated features that make certain people more likely to be bitten by mosquitoes. These studies showed either no difference in biting rates by gender[17,39] or higher biting rates for males compared to females[8,18–20], consistent with our findings. They also observed that individuals aged >15 years tend to receive the most bites from malaria vectors, a pattern often attributed to higher body surface area[8,18–20]. However, we observed that school age boys have higher biting rates than other age groups, and others have observed that biting preferences by age varied between sites and seasons[17]. One potential reason for this is that elements of human attractiveness beyond surface area, such as body odor[5], play a role in host choice. Furthermore, our observation that people sleeping in the same sleeping space were more likely to receive bites from mosquitoes that took multi-source meals also supports the notion that accessibility plays a role in host selection. Taken together, our observations highlight that mosquito biting choices likely result from a combination of host attractiveness, human behavior, and accessibility.

Our study provides a unique opportunity to consider the potential evolutionary advantages resulting from the influence of mosquito behavior by parasites on parasite diversification. Genetic diversification arises from de novo mutation and from recombination during the sexual stages in the mosquito; the latter likely plays a major role in adaptation[40]. For diversifying recombination to occur, the mosquito must be infected with multiple distinct *P. falciparum* strains during the span of a few hours[41]. This can occur if the mosquito feeds on multiple infected individuals in a short timespan (mosquito super-infection),

which occurred only 3.5% of the time in our dataset. Alternatively, polygenomic mosquito infections can also result when a mosquito feeds upon a person who has been co- or super-infected with multiple parasite strains, which is a common phenomenon in high-transmission settings like ours, where we have observed up to 17 strains in human infections[42]. Our observation that infectious mosquitoes appear to be more likely to bite infected individuals provides a plausible mechanism by which these genetically diverse infections accumulate within hosts, and thereby increases the possibility of advantageous recombination in a subsequent mosquito co-infection. This would allow parasites to explore a more diverse fitness landscape through recombination between different strains from different initial infections. Moreover, our observation that *An. gambiae s.s.* and *An. funestus* have relatively similar biting preferences and share haplotypes suggests that there is likely one recombining reservoir of *P. falciparum* strains shared amongst both of these species. Given the evidence that diverse blood-stage infections may be partially prevented by prevalent blood-stage *Plasmodium* infections[43], this phenomenon suggests a mechanism by which the parasite has adapted to generate opportunities for out-crossing and genetic diversification.

Our study has several limitations. First, it was not possible to determine the initial feeding status of reared mosquitoes. Furthermore, the small sample size of reared infected abdomens limited our ability to make inferences about differential onward *P. falciparum* transmission to mosquitoes. However, despite these limitations we were still able to estimate the maximum human-to-mosquito transmission efficiency to be between 0.12–0.22, which aligns with what has been observed in direct- and membrane-feeding experiments[11,44,45]. Second, as mentioned above, parasite detection in mosquito head-thoraces could not distinguish definitively between sporozoite and blood-stage forms; future experimental and observational studies can further explore sporozoite-induced control of feeding behavior. Third, our collections may have incompletely captured all fed mosquitoes. However, mosquitoes that evaded capture are likely a random sample of the overall population and are not expected to be biased toward one specific group of participants. Finally, we could not match all mosquito STR profiles to people indicating some bites may have been on individuals not enrolled in our study; however, our ability to incorporate a portion of incomplete and multisource profiles into our analysis increased our sample size by 37%[46], allowing us to more accurately use the matched bloodmeals and human infection data to estimate potential contributions to the transmission reservoir of different groups of people. Additionally, these estimates assume equal transmission efficiency and gametocyte carriage amongst groups, which several studies have shown are also heterogeneous and nonrandom. Nonetheless, our large sample size of matched bloodmeals representing longitudinal observations of the same participants over an entire transmission season provide robust insights into mosquito biting behaviors and the potential impact on transmission and parasite diversity.

In conclusion, we observed that school-age boys are disproportionately bitten and disproportionately contribute to onward transmission to mosquitoes, and that infectious mosquitoes appear to be more likely to bite infected individuals. Our approach to measuring and quantifying mosquito biting bias in a natural setting contributes to accurate estimation of parasite transmission dynamics, and these observations provide insight into parasite diversity and evolution as well as the development of transmission-reducing interventions.

# Methods

## Ethical statement
We obtained written informed consent from all heads of households and all participants or their parent for those <18 years; the latter also provided verbal assent if >8 years. The study was approved by the ethical review boards of Moi (2017/36) and Duke (Pro00082000) Universities.

## Study population and data collection
We analyzed samples and data collected from July 2020 to September 2021 from a previously described ongoing cohort[47]. Briefly, in Bungoma County, Kenya, all residents above 1 year of age were included from 75 households, which were selected by radial sampling in 5 villages of varying malaria intensity. We collected basic demographic and household data yearly. We collected dried blood spots (DBS) from fingerpricks from all participants every month (active collection) and recorded self-reported bed net use specifically on the night prior to DBS collection. We captured interval episodes of malaria by providing testing by malaria rapid diagnostic test (RDT) upon request owing to self-reported symptoms; those with positive RDTs were referred for treatment with Artemether-Lumefantrine. A DBS was also collected at these visits (passive collection). One morning each week for three weeks each month, we collected indoor-resting mosquitoes from each household using vacuum aspiration with Prokopaks. Mosquitoes were collected twice per month for bloodmeal analysis and once per month for rearing. Mosquitoes collected for rearing were collected on the same day as monthly DBS collections. To simplify our analysis, we assume that we collected all mosquitoes that bit a person in the household that night.

## Entomological methods
Collected mosquitoes either underwent bloodmeal analysis or were reared. For bloodmeal analysis, female *Anopheles spp.* were immediately processed after collection, identified morphologically, and their abdominal status graded as freshly fed, half-gravid, gravid, or unfed. Each was transected to separate the head-thorax, wings, and abdomen. Abdomens graded as bloodfed were pressed onto filter paper, and other parts were stored in tubes.

Mosquitoes collected for rearing were released into household-specific cages inside an insectary maintained at $27 \pm 2°C$ and $80 \pm 10\%$ relative humidity. Mosquitoes fed on cotton saturated with 10% sucrose in distilled water for 7 days prior to sacrificing with chloroform. All female *Anopheles spp.* were morphologically identified, transected, and stored as described above.

## Molecular methods
Primer sequences for all molecular assays are provided in Supplementary Data 1.

**Detection of *P. falciparum* by qPCR.** gDNA was extracted using Chelex-100 from all i) DBS, ii) head-thoraces of mosquitoes processed for bloodmeal analysis, and iii) abdomens of reared mosquitoes. gDNA from each DBS and mosquito abdomen was tested in duplicate in a TaqMan real-time PCR assay targeting the *P. falciparum pfr364* motif as previously described[42]. Samples were defined as *P. falciparum*-positive if: i) both replicates amplified *P. falciparum* with Ct values < 40 or ii) 1 replicate amplified *P. falciparum* with Ct value < 38. Parasite densities were estimated on each reaction plate using standard curves, which consisted of extracts from *P. falciparum* 3D7 parasites across a range of parasites/uL (for DBS) or of *P. falciparum* 3D7 gDNA across a range of ng/uL (for mosquitoes). The latter was converted to parasite density using a genome size of 22.8 Mbps. We assume that *P. falciparum* detected in the head-thorax is predominantly in the sporozoite stage.

**_P. falciparum csp_ amplicon deep sequencing.** All *P. falciparum*-positive DBS and reared mosquito abdomens were genotyped across a variable segment of the parasite circumsporozoite gene (*csp*) as previously described[48] and detailed in the Supplementary Methods.

**Resolution of *Anopheles gambia* sibling species by PCR.** gDNA was extracted by a hotshot technique[49] from a single wing of female *Anopheles* mosquitoes that were morphologically identified as

*An. gambiae s.l.* Sibling species were resolved using a multiplex assay[50] based on the intergenic spacer region (IGS) which gives bands of differing sizes for each of 5 species in the *An. gambiae* complex.

**Human genotyping by short tandem repeats.** We genotyped human blood in both a DBS from each participant and in each mosquito abdomen classified as bloodfed for which a blood spot was available (1064/1563). We extracted gDNA from each sample type using Chelex-100, and genotyped using the Promega Geneprint 10 assay as described in Lapp et al.[46]. This assay does not amplify non-human blood sources.

### Matching mosquitoes and people
**Matching bloodmeals to human hosts.** Mosquito bloodmeals were matched to cohort participants using the bistro R package (v0.2.0)[46], which can provide matches for incomplete STR profiles and bloodmeals derived from multiple human sources. Briefly, matches based on weight-of-evidence likelihood ratios were determined for each mosquito-human pair and individual mosquito-based thresholds. The estimated number of contributors (NOC) to a bloodmeal is computed as *ceiling*[*max*(*a*)/2], where *a* is the number of alleles at a locus. We define single-source bloodmeals as those with NOC = 1 and multi-source bloodmeals as those with NOC > 1.

**Matching infected mosquito abdomens to infected humans.** We matched *P. falciparum* infections in reared *Anopheles* abdomens to people that might have infected them by first subsetting to *P. falciparum*-infected individuals present in the family compound on the day the mosquito was collected. Among these mosquito-human pairs, we define a match as any pair that shared at least one *csp* haplotype.

### Estimating the maximum probability of a mosquito becoming infected after biting an infected individual
If we define $B_{kI}$ as the mosquito biting $k$ infected people, then our goal is to compute the probability that a mosquito was infected given that they bit one infected person, $P(M_I | B_{1I})$. Briefly, we use the equation:

$$P(M_I) = \sum_{k=0}^{n} P(M_I | B_{kI}) P(B_{kI}) \qquad (1)$$

Where $P(M_I)$ is the probability of a mosquito becoming infected. We take $P(M_I | B_{0I})$ to be zero, assuming all infections in the mosquito abdomen had to arise from the most recent biting event. Given that the probability of a mosquito becoming infected after biting $j > 1$ infected people is greater than or equal to that after biting one infected person (i.e. $P(M_I | B_{jI}) \geq P(M_I | B_{1I})$), we assume $P(M_I | B_{jI}) = P(M_I | B_{1I})$ and solve for a maximum transmission efficiency $P(M_I | B_{1I})$:

$$P(M_I | B_{1I}) = \frac{P(M_I)}{P(B_{1I})} \qquad (2)$$

$P(M_I)$ was considered to be the proportion of abdomens from reared mosquitoes that developed *P. falciparum* infections divided by the proportion of immediately processed mosquitoes that recently fed (i.e. freshly fed, half-gravid, or gravid). The probability that a mosquito bit one or more infected people was computed from the STR matching data using only mosquitoes with complete STR profiles. We computed $P(M_I | B_{1I})$ for 1000 bootstrapped replicates of the data and obtained a 95% bootstrapped confidence interval (CIs) by taking the 2.5% and 97.5% quantiles of the bootstrapped values.

### Calculating biting bias
We used the gini function in the R package DescTools v0.99.48[51] to determine the Gini index and 95% CI for mosquito biting bias based on

each person's mean nightly bites, calculated as the cumulative number of bloodmeals matched to an individual divided by the total person-nights in the study. The denominator (person-nights) includes only nights where mosquitoes were collected for immediate processing and the person was physically present in the household at the monthly survey immediately preceding or immediately following the night of collection (IQR: 7–14 days). If a person was not present for a monthly survey within 30 days of that night, we assume that they were not living in the house during that time period. We also assume that the blood matches between freshly fed mosquitoes and people were due to a bite that occurred within the previous 24 hours. We define random biting as each person having the same probability of receiving a mosquito bite per night at risk. To evaluate the expected Gini index under a random distribution of bites, we computed the Gini index for 1000 simulated random bite distributions where the probability of a person being bitten was weighted by the amount of time they were present during the study period and identified the 2.5%, 50%, and 97.5% percentiles.

### Risk factor analysis
We performed a risk factor analysis to assess the characteristics of people who were more likely to be bitten by mosquitoes. For each household-night with at least one STR-matched mosquito, we included in the sample anyone who slept in the household within 30 days of that night; people were considered to have received $n$ bites, where $n$ is the number of STR-typed mosquitoes that they matched to on that night. We used the glmmTMB function from the R package glmmTMB v1.1.6[52] to run a negative binomial model (family = nbinom2) with a random effect on the person. The outcome was the number of bites a person received on a given night. Risk factors included gender, categorized age (<5, 5–15, and >15 years), whether the person slept under a net, and whether the person was infected with *P. falciparum*. Detailed description of risk factors, adjustors, sensitivity analyses, and subsetted models are in the Supplemental Methods.

### Choice model analysis
We computed a weighted discrete choice model for each human risk factor for being bitten using the R package mlogit v1.1.1[53]. We subset to include only *An. gambiae s.s.* and *An. funestus* mosquitoes with STR and sporozoite detection results available. The following three mosquito characteristics were included in each model: mosquito species, single- vs multisource meal, and *P. falciparum* presence in the mosquito head-thorax (i.e. mosquito infectiousness). Each mosquito was identified as an individual, and all people present in the household when that mosquito was collected were included in the model. Values of the human characteristics were the alternatives, weighted by the number of people in the household in each group. An example of the data structure and analysis can be found in the Supplementary Methods. Mosquitoes that matched to two people, and the people present in the household at the time, were included twice in the model, with each bitten person coded as being bitten in one of the entries. We also computed a choice model with mosquito *P. falciparum* load in the head-thorax as a continuous variable; uninfected mosquitoes were assigned a *P. falciparum* density of 0. Finally, we assessed whether the human characteristics associated with taking multisource meals were driven by shared sleeping spaces (See Supplementary Methods).

### Data analysis and visualization
All analyses and visualizations were performed in RStudio v2023.06.1.524[54] with R v4.2.1[55] using the following packages: tidyverse v2.0.0[56], haven v2.5.3[57], Hmisc v5.0.1[58], ggpubr v0.6.0[59], geomtextpath v0.1.1[60], DiagrammeR v1.0.9[61], broom v1.0.5[62], broom.mixed v0.2.9.4[63], ggeffects v1.3.2[64], gtsummary v1.7.0[65], ggmosaic v0.3.3[66], biostat3 v0.1.8[67], modelr v0.1.11[68].

## Reporting summary

Further information on research design is available in the Nature Portfolio Reporting Summary linked to this article.

## Data availability

*Plasmodium falciparum* sequences are available as BioProject PRJNA1064031. Data from human participants in this study are not made available in an open repository due to privacy issues and conditions of IRB approval. Data can be requested from the Principal Investigators (O'Meara and Taylor), who will respond to requests within 2 weeks. Investigators interested in the dataset will be asked to provide a brief study description/analysis plan. Data will be provided via a secure link. No identifying information will be shared and data recipients will not be permitted to share data with other investigators. Source data are provided with this paper.

## Code availability

All code used for analysis and figures is available on Github: https://github.com/duke-malaria-collaboratory/imbibe_manuscript.

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

## Acknowledgements

We thank field technicians Ibrahim Khaoya, Lucas Marango, Ezna Mukeli, Eric Nalianya, Jane Nyongesa, Lilian Nukewa, Edith Wamalwa, and Aggrey Wekesa for their engagement with the study participants; Joseph Kipkoech Kirui, Sarah Laing, Julius Maiyo and Emily Robie for operational assistance and coordination; Kelsey Sumner and Erica Zeno for scientific discussion; Thynn Thane, Jillian Grassia, Jenna Decurzio, Laura-Leigh Rowlette and Scott Langdon for sample processing; and Jamie Mills, Robert Rono, Francis Kithuku and Nikita Poujai for administrative support. Ultimately, we are indebted to the study household members for their participation in this study. This work was supported by NIAID (R01AI146849 to W.P.O. and S.M.T. and F32AI149950 and K01AI175527 to C.F.M.). The content is solely the responsibility of the authors and does not necessarily represent the official views of the National Institutes of Health.

## Author contributions

W.P.O., S.M.T., A.A.O., J.N.M., A.W., and C.F.M. acquired funding. J.N.M., A.A.O., W.P.O., and L.A. performed project administration. E.O., T.C., M.A., E.F., and C.F.M. performed the investigations. E.K. curated the data. C.F.M., Z.L., S.M.T., and W.P.O. conceptualized the project and wrote the original manuscript draft. C.F.M. and Z.L. developed the methodology and software, and performed the formal analysis, and visualization. T.M., A.W., and S.B. supported data analysis. W.P.O. and S.M.T. supervised the project. All authors reviewed and edited the manuscript. W.P.O. and S.M.T. contributed equally.

## Competing interests

The authors declare no competing interests.

## Additional information

**Supplementary information** The online version contains
supplementary material available at

Steve M. Taylor or Wendy Prudhomme O'Meara.

**Peer review information** *Nature Communications* thanks John Keven,
Douglas Paton and the other, anonymous, reviewer(s) for their con-
tribution to the peer review of this work. A peer review file is available.

