## [Peer Review File · Nature Communications]

Plasmodium falciparum infection in humans and mosquitoes influence natural Anopheline biting behavior and transmissionREVIEWER COMMENTS

Reviewer #2 (Remarks to the Author):

In this manuscript, by Markwalter and colleagues, the authors conducted a longitudinal study of Anopheles mosquito blood feeding preferences and Plasmodium falciparum transmission in a malaria-endemic region of Kenya between 2021 and 2022. By repeat sampling of resting Anopheles mosquitoes across 75 households, the authors were able to determine host feeding preference down to the individual level and Plasmodium infection status in vector mosquitoes, and link these data to human factors including age, sex, bed net usage, malaria infection status etc. in order to assess relative risk factors between cohorts.

This is a striking, well-written and well-presented manuscript describing a complex study that has been carried out with a commendable degree of rigor. The most important findings of this study; 1) that in this setting school age boys receive a disproportionate number of bites, and are responsible for the plurality of onward transmission of Plasmodium to mosquitoes – and thus to subsequent human hosts, and 2) that, specifically, infectious *An. gambiae*, and *An. funestus* harboring sporozoites in their salivary glands strongly prefer to feed on malaria infected human hosts, are genuinely impactful with important consequences for both the control and elimination of malaria, as well as the reproductive and evolutionary biology of this parasite. Overall, I have no reservations in recommending this manuscript be accepted for publication pending a response to the minor points raised below.

Doug Paton,
Assistant Professor,
College of Veterinary Medicine,
University of Georgia

Questions, Comments, Points to address:

It would be interesting to know what the absolute mosquito species distribution was on sample days across the study? Did the relative densities of *An. gambiae* and *An. funestus* change over time? Was one species more dominant? A sentence addressing this in the results or discussion is recommended.

I was surprised at the number of bites school age, Pf positive, boys were receiving even when under nets (Figure 2C). While clearly a result of strong preference for feeding on this cohort borne out by this study, how much does insecticide resistance factor into this? Two recent studies show that both *An. gambiae* and *An. funestus* populations in Western Kenya (around Kisumu and Kakamega, PMID: 38464038, PMID: 33647049) exhibit significant resistance to the pyrethroid insecticides used in bed nets.

Related: Although present in the manuscript indirectly in Figure 2C, some discussion of absolute bed net usage across age/sex cohorts would be of interest. Of note it appears from this data that Male 5-15 y.o. children are the only cohort that predominantly do not sleep under nets.

Line 314: "abdominal status". Here and elsewhere I suggest using "infection" rather than "abdominal", as the current wording is a little confusing. Further, while the qPCR technique used for parasite detection is a sensitive and appropriate tool, imaging and sizing midgut oocysts may have provided some limited insight into the prior infection status, based on relative oocyst size and morphology, whilst still allowing downstream molecular analysis.

Is it possible that the unmatched STR profiles were from adjacent animal blood sources? Would the kit used for this analysis amplify non-human blood sources?

Reviewer #3 (Remarks to the Author):

I reviewed this manuscript with great interest as this work falls within my area of expertise and overlaps with my own research. In this paper, Markwalter and colleagues investigated the frequency distribution of malaria mosquito bites among humans and how the distribution pattern affects malaria transmission in five villages in Western Kenya. They approached this problem by collecting indoor-resting Anopheles mosquitoes and linking them to the person they fed on by matching human genetic profiles analyzed from the mosquito blood meals to those from blood samples of the village residents with known sex, age and Plasmodium falciparum infection status. They also collected data on P. falciparum infection status of the mosquitoes.

Two key findings were presented in this paper. The first was that the distribution of mosquito bites among humans was extremely clustered (i.e., non-random) with a small proportion (20%) of the village residents contributing to most (86%) of the mosquito blood meals and this bias blood feeding behavior was in favor of school-age males. Coupled with the understanding that school-age children harbor chronic asymptomatic P. falciparum infections and were the most infectious age group to mosquitoes in feeding assays, they concluded that members of this human demographic are the main drivers of P. falciparum transmission. Stated differently, school-age boys disproportionately contribute to onward transmission (human → mosquito) of P. falciparum. The second finding was that compared to uninfected mosquitoes, infected mosquitoes (those that carry sporozoites) were more likely to bite infected than uninfected humans, which led them to conclude that P. falciparum sporozoites manipulate mosquito feeding behavior in favor of infected people.

Regarding the first finding, the data were sufficient and the statistical or quantitative methods used were, in my opinion, appropriate and robust. The non-random biting was consistent with the results of previous works by other groups including my own group. The identification of school-age male humans as the potential driver of malaria transmission is an important new finding, one that is consistent with my own team's preliminary data from a similar study also in Western Kenya. From most reviewers' standpoint, the convergence of results from independent researchers adds more confidence to this conclusion. This finding contributes significantly to the field of vector biology and our understanding of malaria transmission and warrants publication in a journal like Nature Communications.

However, for the second finding, while the statistical methods used were appropriate for the data, I am skeptical about the data itself and hence the observation and conclusion that were inferred from it. The inference that P. falciparum sporozoites manipulate mosquito feeding behavior to favor infected people hinges on the assumption that the Plasmodium DNA detected by PCR in the head-thorax of the blood-fed mosquitoes were from the sporozoite stage of the parasites. However, the PCR method used was not specific to the sporozoite stage, which means that human stages of the parasite acquired with the blood meals could potentially be detected and erroneously scored as sporozoite infections. Compared to mosquitoes that bite uninfected people, those that bite infected people are more likely to pick up human stages of the parasites with the blood meal and consequently are more likely to be tested positive in the PCR. This would affect the data by inflating the "sporozoite" infection rate in this group of mosquitoes, eventually resulting in an erroneous conclusion such as the inference that sporozoites manipulate mosquitoes to feed on infected people. Looking at it in reverse, people who are fed upon excessively (e.g., school-age boys) have higher chances of being inoculated and are more likely to be infected than those who are fed upon less frequently and therefore are likely to pass on Plasmodium-positive blood meals to their biters, creating the misleading result described above.

Restricting the PCR test to the head-thorax does not eliminate this problem fully or certainly as human stages of the parasites can be trapped in the foregut which passes through the head-thorax of the insect. Without addressing this potential flaw in the method, the authors cannot confidently conclude that sporozoites manipulate the mosquito biting behavior to favor infected people. Methods like the sporozoite enzyme-linked immunosorbent assay (ELISA) or RNA-based PCR are sporozoite-specific. One way to address this problem is to apply these sporozoite-specific tests on the head-thorax of the two groups of blood-fed mosquitoes (those that fed on uninfected vs. infected people) to confirm if the sporozoite rates is statistically higher in those that fed on infected people. Another way is to reference a study that has demonstrated beyond reasonable

doubt that PCR tests of head-thorax of blood-fed mosquitoes does not inflate sporozoite infection rates. If the authors cannot meet these or such suggestions, they should acknowledge this potential flaw in their work and present the observation as a hypothesis that requires further testing and not as a factual conclusion.

I wanted to conclude by saying that I presented my critique of the work while acknowledging my own potential ignorance of some aspects of this subject. Nevertheless, I hope that my comments help the authors improve their manuscript. I look forward to their rebuttal of my comments.

Reviewer #4 (Remarks to the Author):

This manuscript describes analyses of data collected from a 15 month study in Western Kenya. Their main findings are that school-age boys are disproportionately bitten and disproportionately contribute to onward transmission to mosquitoes and that infectious mosquitoes are more likely to bite infected individuals. This is a nicely written narrative and highlights its novelty well. My main comments are about clarity in the methods section, which I think could help the readers understand the results better.

Methods:

- 1) p18 - What method was used to assess bednet usage? Seeing the net / asking the participants if they used one? What question was used? Slept under bednet last night or more generally?
- 2) p 18/19- What is the specificity of the RDT/ PCR assays?
- 3) p21 - I think explaining the conditional probability notation earlier could help a reader who isn't familiar.
- 4) p22 - i think there is a typo "after biting two infected people"
- 5) p22- would be helpful to see the exact equation computed rather than giving the unsimplified one.
- 6) p22 how was bootstrapping done
- 7) p22 how did you address that might not have captured all the mosquitoes that bit a person that night?
- 8) p22 confused how nights in the study are defined? Mosquitoes that bit people on different days of the week to mosquito collection wouldn't necessarily show up in the study.
- 9) p23 justification why include people who stayed in the house within 30 days.

Results

- 10) Were results consistent between different seasons?
- 11) p7 - how did you account that some of the mosquitoes that you reared may have bitten humans in your sample population too.
- 12) p8 - confused how know the bites are from a single night. Assume all blood fed mosquitoes bit someone the previous night? Is this a fair assumption (am not an expert).
- 13) p8 - it would be interesting to know the variance / range of estimated number of bites per month per person (estimated 12 bites per person per month).

Response to reviewers:

Reviewer 2

1. *It would be interesting to know what the absolute mosquito species distribution was on sample days across the study? Did the relative densities of *An. gambiae* and *An. funestus* change over time? Was one species more dominant? A sentence addressing this in the results or discussion is recommended.*

RESPONSE: This information was hidden in Figure S1A. We have added text to the first paragraph of the results explicitly describing what is shown: “Of the 3502 mosquitoes for which we could assign species, most were *An. gambiae* (1611 *An. gambiae* s.s., 46%; 130 *An. gambiae* s.l., 4%) or *An. funestus* (1600, 46%). In 2021, *An. gambiae* s.s. counts peaked slightly before *An. funestus* counts (**Figure S1A**).”

2. *I was surprised at the number of bites school age, Pf positive, boys were receiving even when under nets (Figure 2C). While clearly a result of strong preference for feeding on this cohort borne out by this study, how much does insecticide resistance factor into this?*

RESPONSE: This is a good point; we are currently investigating insecticide resistance in another study. Insecticide resistance might reduce the effectiveness of ITNs in preventing bites across all individuals using nets, but not school age boys specifically. It is more likely that this group is not protected by a net during times that anopheles are biting (for example, outside of the net late at night or failure to open and use the net consistently). We are exploring insecticide resistance in local mosquito populations in follow-on studies, but there is no evidence to suggest that insecticide resistance might influence biting differentially across demographic groups.

3. *Although present in the manuscript indirectly in Figure 2C, some discussion of absolute bed net usage across age/sex cohorts would be of interest. Of note it appears from this data that Male 5-15 y.o. children are the only cohort that predominantly do not sleep under nets.*

RESPONSE: We have added two panels to Figure S2 depicting the proportion of monthly visits from active case detection where an individual was infected with *P. falciparum* (B) and reported sleeping under a bed net (C) (see below). Although school age male children did have the lowest net usage, net use among older males and among school age female children was also substantially lower than that of children under 5 years (male or female) and females >15 years. We have also added corresponding text to the results: “**Individuals**

were infected at a median of 20% of monthly visits, with little variation across age and gender (**Figure S2B**). However, bed net use did vary, with males aged 5-15 reporting the lowest bed net use (**Figure S2C**)."

Figure S2: Overview of human infections and demographics during the study period.

(A) Human infections over time for active and passive case detection. (B) Proportion of monthly visits from active case detection that an individual was infected with *P. falciparum*, by age and gender. Kruskal-Wallis p-values between ages within a gender are shown on the plot. Kruskal-Wallis p-values between genders within age groups were 0.93 (<5 years), 0.82 (5-15 years), and 0.03 (>15 years). (C) Proportion of monthly visits from active case detection where an individual reported sleeping under a bed net, by age and gender. Kruskal-Wallis p-values between ages within a gender are shown on the plot. Kruskal-Wallis p-values between genders within age groups were 0.7 (<5 years), 0.02 (5-15 years), and 0.08 (>15 years).

4. *Line 314: “abdominal status”. Here and elsewhere I suggest using “infection” rather than “abdominal”, as the current wording is a little confusing. Further, while the qPCR technique used for parasite detection is a sensitive and appropriate tool, imaging and sizing midgut oocysts may have provided some limited insight into the prior infection status, based on relative oocyst size and morphology, whilst still allowing downstream molecular analysis.*

RESPONSE: We apologize for the confusion – abdominal status is not related to infection here, it is related to whether the mosquito was freshly fed, half-gravid, or gravid. We have clarified this in the text:

Results: “Most mosquitoes (3038) were immediately processed to determine abdominal status (freshly fed, half-gravid, gravid, or unfed) and preserve the blood from freshly fed mosquitoes (1563) to match to cohort members.”

Methods: “For bloodmeal analysis, female *Anopheles spp.* were immediately processed after collection, identified morphologically, and their abdominal status graded as freshly fed, half-gravid, gravid, or unfed.”

5. *Is it possible that the unmatched STR profiles were from adjacent animal blood sources? Would the kit used for this analysis amplify non-human blood sources?*

RESPONSE: The STR-typing method that we used does not amplify non-human blood sources. Therefore, the unmatched STR profiles were either from people who were not included in the study or were too incomplete to unequivocally match to a cohort individual. The freshly fed mosquitoes for which we did not obtain profiles may have been from non-human blood sources. We have added this sentence to the methods to clarify: “This assay does not amplify non-human blood sources.” We have also interpreted this in the limitations: “...indicating some bites may have been on individuals not enrolled in our study”

Reviewer 3

6. *The inference that *P. falciparum* sporozoites manipulate mosquito feeding behavior to favor infected people hinges on the assumption that the Plasmodium DNA detected by PCR in the head-thorax of the blood-fed mosquitoes were from the sporozoite stage of the parasites. However, the PCR method used was not specific to the sporozoite stage, which means that human stages of the parasite acquired with the blood meals could potentially be detected and erroneously scored as sporozoite infections....Restricting the PCR test to the head-thorax does not eliminate this problem fully or certainly as human stages of the parasites can be trapped in the foregut which passes through the head-thorax of the insect. Without addressing this potential flaw in the method, the authors cannot confidently conclude that sporozoites manipulate the mosquito biting behavior to favor infected people.*

RESPONSE: This is a good point. We have added our assumption explicitly to the methods: "We assume that *P. falciparum* detected in the head-thorax is predominantly in the sporozoite stage."

To test this assumption, we have added a sensitivity analysis to the supplementary materials of the manuscript. Given the small volume of the foregut and high prevalence of low-density asymptomatic infections in our community members, we expect that the contribution is very low of human stage parasites to the measured *P. falciparum* loads in the mosquito head-thoraces. Therefore, the risk of misclassifying mosquitoes as sporozoite-positive is highest in the head-thoraces with the lowest parasite loads. Thus, for our sensitivity analysis, we re-evaluate the main discrete choice analysis with increasing *P. falciparum* density thresholds for defining mosquitoes as sporozoite-positive. As shown below in **Table S4**, the relative risk ratio of infectious mosquitoes biting infected individuals remains high and statistically significant as mosquitoes with low-density head *P. falciparum* loads are reclassified as *Pf*-negative, supporting our interpretation that the effects we observe can be attributed to sporozoite infection in the mosquito head-thoraces. Changes to the text and supplement are below:

Results: "This association could result from inherent biases for mosquitoes to bite specific people, or from the presence of non-sporozoite stage *P. falciparum* from a bitten infected person being detected in the head-thorax. Based on the assumption that the contribution of human stage parasites to the measured *P. falciparum* loads in the mosquito head-thoraces would be low, we conducted a sensitivity analysis that tested four different thresholds to classify mosquito head-thoraces with low *P. falciparum* loads as sporozoite-negative. The value of threshold did not change the direction of the significance of the observed effect. To

further assess the association between the presence of sporozoites and a preference to bite infected people, we next”

Table S4: Sensitivity analysis increasing the *P. falciparum* density threshold for defining a mosquito-head thorax as sporozoite-positive.

Mosquito feature	Sporozoite load threshold (genomes)	Relative Risk Ratio (95% CI)
		Human P. falciparum infection: positive vs. negative (ref)
Sporozoite positive vs. negative (ref)	0	2.76 (1.65, 4.61)
Sporozoite positive vs. negative (ref)	10	3.37 (1.82, 6.25)
Sporozoite positive vs. negative (ref)	50	2.71 (1.42, 5.19)
Sporozoite positive vs. negative (ref)	100	3.25 (1.42, 7.43)
Sporozoite positive vs. negative (ref)	200	20.26 (1.57, 65.70)

Finally, we toned down the wording of our conclusions throughout the manuscript:

Results: “This observation further supports the notion that the preference of Anopheline mosquitoes for biting people infected with *P. falciparum* is enhanced by the presence of sporozoites in the mosquito.”

Results: “Taken together, these results suggest that school-age boys disproportionately contribute to onward *P. falciparum* transmission and that *P. falciparum* sporozoites may modify mosquito biting preferences to favor feeding on infected people.”

Discussion: However, the PCR method we used to detect *P. falciparum* in mosquito head-thoraces cannot discriminate between different stages of the parasite; therefore, it important that further studies investigate whether this observation holds when specifically testing for sporozoite-stage *P. falciparum* in the mosquito head-thorax.”

Discussion: “Either way, our observations support the notion that, in this natural setting, *P. falciparum* parasites may manipulate Anopheline vectors to favor feeding on humans who are already likewise infected with *P. falciparum*.”

Discussion: “Our observation that infectious mosquitoes appear to be more likely to bite infected individuals...”

Discussion: “...as mentioned above, parasite detection in mosquito head-thoraces could not distinguish definitively between sporozoite and blood-stage forms; future experimental

and observational studies can further explore sporozoite-induced control of feeding behavior”.

Discussion: “...infectious mosquitoes appear to be more likely to bite infected individuals..”

Reviewer 4

7. *Methods p18 - What method was used to assess bednet usage? Seeing the net / asking the participants if they used one? What question was used? Slept under bednet last night or more generally?*

RESPONSE: We have added this clarification to the methods: “We collected dried blood spots (DBS) from fingerpricks from all participants every month (active collection) and recorded self-reported bed net use specifically on the night prior to DBS collection.”

8. *Methods p 18/19- What is the specificity of the RDT/ PCR assays?*

RESPONSE: We used Carestart Pf HRP2 RDTs and SD Bioline Pf HRP2 RDTs, both of which have a specificity of >99%. The PCR assay amplifies a *P-falciparum*-specific motif and does not amplify other Plasmodium spp.

9. *Methods p21 - I think explaining the conditional probability notation earlier could help a reader who isn't familiar.*

RESPONSE: We have added this to the methods: “If we define B_{kI} as the mosquito biting k infected people, then our goal is to compute the probability that a mosquito was infected given that they bit one infected person, $P(M_I|B_{1I})$.”

10. *Methods p22 - i think there is a typo "after biting two infected people”*

RESPONSE: We have changed the text here.

11. *Methods p22- would be helpful to see the exact equation computed rather than giving the unsimplified one.*

RESPONSE: We have updated the methods to make this part clearer:

“Given that the probability of a mosquito becoming infected after biting $j > 1$ infected people is greater than or equal to that after biting one infected person (i.e. $P(M_I|B_{jI}) \geq$

$P(M_i|B_{1i})$), we assume $P(M_i|B_{ji}) = P(M_i|B_{1i})$ and solve for a maximum transmission efficiency $P(M_i|B_{1i})$:

$$\underline{P(M_i|B_{1i}) = P(M_i)/P(B_{1i})}$$

12. *Methods p22 how was bootstrapping done*

RESPONSE: We have added more information about this to the methods: “We computed $P(M_i|B_{1i})$ for 1000 bootstrapped replicates of the data and obtained 95% bootstrapped confidence intervals (CIs) by taking the 2.5% and 97.5% quantiles of the bootstrapped values.” We also realized that the original CIs were computed with only 100 bootstrapped replicates; we re-computed the values with 1000 replicates and updated the corresponding values in the text.

13. *Methods p22 how did you address that might not have captured all the mosquitoes that bit a person that night?*

RESPONSE: Since this question is relevant for multiple analyses that we did, we added this sentence to the “Study population and data collection” section of the methods to clarify: “To simplify our analysis, we assume that we collected all mosquitoes that bit a person in the household that night. We also added it as a limitation to the discussion: “Second, our collections may have incompletely captured all fed mosquitos. However, mosquitoes that evaded capture are likely a random sample of the overall population and are not expected to be biased towards one specific group of participants.””

14. *Methods p22 confused how nights in the study are defined? Mosquitoes that bit people on different days of the week to mosquito collection wouldn't necessarily show up in the study.*

RESPONSE: We only include nights where mosquitoes were collected in our calculations. We have clarified this in the methods: “The denominator (person-nights) includes only nights where mosquitoes were collected for immediate processing and the person was physically present in the household the monthly survey immediately preceding or immediately following the night of collection (IQR: 7-14 days).”

15. *Methods p23 justification why include people who stayed in the house within 30 days.*

RESPONSE: This is the resolution of the data we have – monthly surveys that indicate whether or not each person was present in the house the day of the survey. We have clarified this in the methods: “If a person was not present for a monthly survey within 30

days of that night, we assume that they were not living in the house during that time period.”

16. *Results Were results consistent between different seasons?*

RESPONSE: Unfortunately, we do not have enough data to determine whether results were different between different seasons; our study spanned 15 months, and we observed very few matched bloodmeals during the low season. We have added a sentence to the results to highlight this: “Most mosquitoes (3051, 83%) were collected during the high-transmission season in 2021 (March-August)”

17. *Results p7 - how did you account that some of the mosquitoes that you reared may have bitten humans in your sample population too.*

RESPONSE: For the biting analyses, we only considered nights on which mosquitoes were collected and immediately processed. Therefore, collections for which mosquitoes were reared were not included in the calculation of biting rates or the denominator of person-nights. We have clarified this in our definition of person-nights (see #15): “The denominator (person-nights) includes only nights where mosquitoes were collected for immediate processing and the person was present in the household for a monthly survey within 30 days of that night.”

18. *Results p8 - confused how know the bites are from a single night. Assume all blood fed mosquitoes bit someone the previous night? Is this a fair assumption (am not an expert).*

RESPONSE: Yes, we assume that all bloodfed mosquitoes bit someone the previous night. We have clarified this in the methods: “We also assume that the blood matches between freshly fed mosquitoes and people were due to a bite that occurred within the previous 24 hours.” We believe this is a reasonable assumption because: (1) we used only fully bloodfed mosquitoes that had not begun to digest the blood meal (usually apparent within the first 8-12 hours after a meal) (Hocking and MacInnes, *Bulletin of Entomological Research* 2009), and (2) complete bloodmeal STR profiles can only be obtained until about 24 hours post-bite, and partial only until about 36 hours post-bite (Ahmed *et al. Insects* 2023) .

19. *Results p8 - it would be interesting to know the variance / range of estimated number of bites per month per person (estimated 12 bites per person per month).*

RESPONSE: We have added this to the results: "...with an estimated 13.2 bites per month per person (range: 0-210; **Figure 2C**).” We have also updated Figure 2C to include the range for each group:

(C) Estimated monthly bites per person (from observed average nightly bites; color and number) by 4 risk factors (net use, *P. falciparum* infection status, gender, and age). n indicates the number of people included present in the group and the bottom number is the range.

REVIEWERS' COMMENTS

Reviewer #2 (Remarks to the Author):

Thank you for your response to my previous comments. I have nothing further to add and am happy for the manuscript to proceed.

Reviewer #3 (Remarks to the Author):

The authors have satisfactorily addressed my concerns in the revised version of their manuscript. I have no hesitations in having their manuscript published asap.

Reviewer #4 (Remarks to the Author):

The authors have thoroughly responded to my comments - thanks. I have nothing further to add